# Application of Viral Vectors for Vaccine Development with a Special Emphasis on COVID-19

**DOI:** 10.3390/v12111324

**Published:** 2020-11-18

**Authors:** Kenneth Lundstrom

**Affiliations:** PanTherapeutics, CH1095 Lutry, Switzerland; lundstromkenneth@gmail.com; Tel.: +41-79-776-6351

**Keywords:** viral vaccines, infectious diseases, cancers, COVID-19 vaccines, self-replicating RNA vectors, DNA-based vaccines, RNA-based vaccines

## Abstract

Viral vectors can generate high levels of recombinant protein expression providing the basis for modern vaccine development. A large number of different viral vector expression systems have been utilized for targeting viral surface proteins and tumor-associated antigens. Immunization studies in preclinical animal models have evaluated the elicited humoral and cellular responses and the possible protection against challenges with lethal doses of infectious pathogens or tumor cells. Several vaccine candidates for both infectious diseases and various cancers have been subjected to a number of clinical trials. Human immunization trials have confirmed safe application of viral vectors, generation of neutralizing antibodies and protection against challenges with lethal doses. A special emphasis is placed on COVID-19 vaccines based on viral vectors. Likewise, the flexibility and advantages of applying viral particles, RNA replicons and DNA replicon vectors of self-replicating RNA viruses for vaccine development are presented.

## 1. Introduction

The recent coronavirus pandemic (COVID-19) has underlined the importance of vaccine development. It has also become clear to the general public that a number of competing approaches for vaccine candidates need to be developed in parallel to achieve success in the shortest possible time. The same strategy should be applied to any vaccine target albeit the global concern related to COVID-19 has drained resources from other important vaccine development initiatives. It should also be pointed out that vaccine development is not restricted to infectious diseases as quite a few approaches have focused on cancer vaccines as discussed below.

The traditional approach, which is still valid and plays an important role in COVID-19 vaccine development against viral infections, relates to the application of killed and live-attenuated vaccines [1]. Moreover, protein subunit and peptide vaccines have become popular, not least due to the development of efficient recombinant protein expression systems in the 1980s and 1990s [2]. The topic of this review is the utilization of viral vectors for vaccine development. In this context, a variety of viral expression systems have been engineered. Typically, expression vectors have been constructed for adenoviruses (Ads), alphaviruses, flaviviruses, measles viruses (MVs), rhabdoviruses, retroviruses (RVs), lentiviruses (LVs), and poxviruses [3,4]. Briefly, Ad vectors are non-enveloped double-stranded DNA (dsDNA) viruses with a packaging capacity of 7.5 kb foreign DNA providing transient episomal expression in a broad range of host cells [5]. Alphavirus- and flavivirus-based vectors are enveloped single-stranded RNA (ssRNA) viruses with a positive polarity, characterized for their self-replicating RNA property, which provides substantial amplification of foreign mRNA directly in infected host cells [6,7]. In contrast, MVs [8] and rhabdoviruses [9] possess an ssRNA genome of negative polarity, which requires reverse genetics to establish appropriate expression vectors. Among these self-amplifying RNA viral vectors, alphaviruses hold a packaging capacity of 8 kb of foreign genes, whereas for the others it is about 6 kb.RVs are ssRNA viruses, characterized by reverse transcription of their genome into DNA, which can be integrated into the host cell genome providing long-term transgene expression [10]. The chromosomal integration of RVs has posed some safety issues especially for gene therapy applications, where insertions in active oncogene loci has triggered the development of leukemia in patients with X-linked severe acute immunodeficiency (SCID-X1) [11]. However, this issue has been addressed by the engineering of self-inactivating RV vectors with targeted integration. Another issue with classic RVs is their inability to transduce non-diving cells. For this reason, many gene therapy and vaccine development activities have switched to LVs, also belonging to the genus of RVs, which otherwise provide the same properties as classic RVs including packaging of up to 8 kb of foreign sequences, but are able to infect both dividing and non-dividing cells [12]. Moreover, integration-defective LV vectors have been engineered based on targeted recombinase-mediated cassette exchange to provide safe episomal status [13]. Poxviruses are large dsDNA viruses with a packaging capacity of over 30 kb of foreign DNA, which have been frequently used for vaccine development [14]. Moreover, the small ssRNA Picornaviruses—especially coxsackieviruses—with the potential to insert 6 kb of foreign nucleic acids, have been engineered as expression vectors [15].

The application of different viral vector systems for vaccine development is reviewed below. The approaches of vaccine development for infectious diseases and cancer are presented in separate sections. Moreover, the accelerated efforts of virus-based vaccine development against COVID-19 are addressed in another section. Although viral vector-based vaccine development has in general relied on the expression of viral surface antigens and tumor-associated antigens for immunization, oncolytic viruses and viral vectors carrying reporter genes have been included in this review due to their capacity of tumor-specific replication, which can provide therapeutic activity similar to what has been discovered for viral vector-based vaccines.

## 2. Viral Vaccines for Infectious Diseases

A common strategy for vaccine development against infectious agents, mainly viruses, has been to introduce immunogenic full-length or truncated viral surface proteins into viral expression vectors for verification of antigen expression in vitro, followed by immunization studies in animal models to evaluate immune responses and potential protection against challenges with lethal doses of pathogenic infectious agents [16]. Due to the large number of preclinical and clinical vaccine studies using viral vectors, it is only possible to present some examples below, with a summary provided in Table 1. Moreover, the main focus is on viral diseases and although vaccines against other types of pathogens have been developed, these are only briefly described at the end of the section.

Although alphaviruses have been frequently used as vaccine vectors, some members of the family, such as Chikungunya virus (CHIKV), have been responsible for severe epidemics in the Republic of Congo [17] and in Reunion [18]. In this context, a chimeric vesicular stomatitis virus (VSV) vector was engineered to express the CHIKV envelope polyprotein (E3-E2-6K-E1) and the Zika virus (ZIKV) membrane-envelope protein (ME) [19]. A single immunization of mice with 1 × 10^7^ pfu induced neutralizing antibodies and resulted in protection against challenges with both CHIKV and ZIKV. In another approach, an Ad-based vaccine strategy was applied for the expression of the Venezuelan equine encephalitis virus (VEE) structural proteins (E3-E2-6K) [20]. Improved codon usage showed a 10-fold increase in antibody responses in BALB/c mice, which also increased protection against challenges with VEE. Moreover, VEE, western equine encephalitis virus (WEE) and eastern equine encephalitis virus (EEE) have been targeted for vaccine development [21]. In this context, vectors for VEE, WEE and EEE have been engineered by removing the furin cleavage site between the E2 and E3 envelope proteins to prevent cleavage of the p62 precursor, which in turn will restrict formation of infectious particles and instead generate virus-like particles (VLPs) [22]. Immunization of mice with 1 × 10^7^ IU of the VEE/WEE/EEE combination or individual VLPs elicited strong neutralizing antibody responses and provided protection against subcutaneous or aerosol challenges with VEE, WEE and EEE [22]. The VEE/WEE/EEE combination of 2 × 10^8^ IU elicited robust neutralizing antibody responses in cynomolgus macaques and showed protection against challenges with VEE and EEE. However, the antibody response against WEE was poor, which also reflected the weak protection seen against WEE challenges. In another approach, the attenuated VEE V4020 strain was administered as a layered DNA/RNA vector into BALB/c mice resulting in a high titer of neutralizing antibodies and protection against challenges with wild-type VEE [23]. Moreover, intramuscular immunization of cynomolgus macaques with the VEE vaccine provided protection against aerosol challenges with wild-type VEE [24].

Related to arenavirus vaccines, Lassa virus (LASV) has been targeted by VSV-based expression of LASV glycoprotein (GPC) [25]. Protection against challenges with LASV strains from Liberia, Mali and Nigeria was obtained in guinea pigs and macaques vaccinated with 1 × 10^6^ and 6 × 10^7^ pfu, respectively. The engineering of an LASV-based replicon system, where the LASV GPC was supplied by Vero cell expression, provided protection in guinea pigs immunized with 5 × 10^5^ focus forming units (ffu) [26]. Similarly, immunization of guinea pigs with 1 × 10^10^ pfu of Ad5-LASV-GPC and Ad5-LASV-NP vaccine candidates demonstrated protection against challenges with lethal doses of LASV [27]. Additionally, an MV-GPC vaccine also provided protection against LASV challenges after a single immunization with 6 × 10^6^ pfu in macaques [28], which supported the initiation of a randomized, placebo-controlled, dose-finding phase I clinical trial in healthy volunteers [29]. Related to other filoviruses, VEE-based expression of Junin virus (JUNV) GPC and Machupo virus (MACV) GPC, respectively, induced humoral immune responses and provided protection in guinea pigs immunized with 1 × 10^7^ pfu [30].

Ebola virus (EBOV), a member of filoviruses, has been an important target for vaccine development due to several Ebola virus disease (EVD) outbreaks, the most recent in 2014–2016 [31]. For instance, the flavivirus Kunjin virus (KUN) was utilized for the expression of the mutant EBOV glycoprotein GP/D637L, which displayed superior cleavability and shedding of GP compared to wild-type GP [32]. Subcutaneous administration of two doses of 1 × 10^9^ KUN-GP/D637L VLPs provided protection in three out of four vaccinated primates. Moreover, immunization with 5 × 10^7^ pfu of VSV-EBOV GP esulted in protection in macaques against challenges with the EBOV-Makona strain [33] and the Zaire strain (ZEBOV) [34]. Similarly, immunization of non-human primates with 1 × 10^12^ pfu of Ad5-EBOV-GP vaccine provided protection against lethal challenges with EBOV [35]. In another approach, a chimeric parainfluenza virus type 3 (HPIV3) with an EBOV-GP envelope showed strong immune responses in guinea pigs immunized with a single intranasal dose and protected them against challenges with guinea pig-adapted EBOV [36]. Due to the success from preclinical studies and the urgent needs for a functional vaccine in humans, VSV particles expressing the EBOV-GP from the Zaire strain (VSV-ZEBOV) were subjected to an open-label, cluster ring vaccination phase III trial [37]. In the trial, 4123 individuals with suspected EVD were immediately vaccinated, while 3528 participants received a delayed vaccination. There were no EVD cases discovered in the immediate vaccination group and only 16 EVD confirmed in the delayed vaccination group indicating that the immunization was efficient. Similar results were obtained from another phase III trial, where 2119 and 2041 participants received immediate and 21 days delayed vaccination, respectively [38]. The vaccination was efficient as no new EVD cases were recorded 10 days after the start of the trial. Related to other filoviruses such as Marburg virus (MARV), immunization of nonhuman primates with 1 × 10^7^ pfu of VSV-MARV-GP particles resulted in protection against challenges with MARV [34]. Similarly, a single intramuscular injection of 1 × 10^10^ ffu of the VEE-based Sudan virus (SUDV) vaccine (VEE-SUDV-GP) provided complete protection in cynomolgus macaques [39]. Interestingly, VEE-SUDV-GP immunization also provided partial protection against challenges with EBOV. However, co-immunization with VEE-SUDV-GP and VEE-EBOV-GP resulted in complete protection against both SUDV and EBOV.

In addition to providing expression systems such as the one based on KUN [7], flaviviruses are known pathogens such as Dengue virus (DENV) and ZIKV causing diseases such a Dengue fever and Zika virus disease, respectively. In attempts to develop a vaccine against Dengue fever, VEE particles expressing the ectodomain of the DENV envelope protein E85 were subjected to a single injection of mice, which resulted in protective immunity against DENV challenges in BALB/c mice [40]. In another approach, administration of 2 × 10^6^ pfu of MV-based vector expressing the DENV domain III of the envelope protein (ED3) to mice, induced DENV-specific immune responses and partial protection against DENV challenges [41]. In the context of clinical trials, DENV vaccine candidates have consisted of live-attenuated vaccines, the chimeric live-attenuated yellow fever-dengue virus tetravalent (CYF-TDV) vaccine or the DENV subunit (DEN-80E) vaccine produced in *Drosophila melanogaster* cells [42]. In the case of ZIKV, a VEE-based replicon RNA expressing the codon-optimized ZIKV *prM* and *E* genes was administered in nanostructured lipid carriers (NLCs) to C57BL/6 mice [43]. It was demonstrated that a single dose as low as 10 ng of the RNA replicon completely protected mice against challenges with ZIKV. As described above, co-expression of a CHIKV polyprotein and the ZIKV ME provided protection in immunized mice [19]. Most of the ZIKV clinical trials conducted relate to live-attenuated or DNA-based vaccines [44]. Although based on an attenuated DENV strain, the 2AA30 vaccine containing the ZIKV ME proteins showed good safety and ZIKV-specific neutralizing antibody responses in a phase I trial in 20 healthy volunteers [45].

Related to hepatotropic viruses, Ad7-based expression of hepatitis B virus (HBV) core antigen (HBcAg) and surface antigen (HBsAg) showed HBV-specific antibody responses in immunized dogs [46]. MV vectors have also been applied for the expression of HBsAg, which showed protection in 50% of rhesus monkeys immunized with 1 × 10^3^ TCID_50_ [47]. Moreover, Semliki Forest virus (SFV), an alphavirus, packaged into a VSV G envelope was used for the expression of the HBV middle surface envelope glycoprotein (MHB) and HBcAg [48]. Immunization of mice with 1 × 10^7^ pfu demonstrated protection against HBV challenges for the SFV-G-MHB vaccine candidate, but not for the SFV-G-HBcAg. In the case of clinical trials, DNA-, live vector-, peptide-based vaccines and cell-based therapies have been preferred to viral-based vaccines [49], although a phase I trial has been initiated for an Ad5 vector expressing a fusion protein composed of a truncated HBV core, a modified HBV polymerase and two HBV envelope domains [50].

The annual influenza virus outbreaks have stressed the importance of the development of effective vaccines. In this context, Ad-based influenza A virus vaccines have been engineered by expression of different portions of the hemagglutinin (HA) protein [51]. BALB/c mice immunized with 5 × 10^10^ Ad particles expressing the full-length HA were protected from challenges with the lethal VN/1203/04 H5N1 influenza A virus strain. Similarly, a single subcutaneous immunization of 5 × 10^10^ Ad particles provided complete protection in chickens. In another approach, VEE particles expressing the HA gene from the Hong Kong influenza A virus isolate (A/HK/156/97) was evaluated in chicken [52]. A single dose of 1 × 10^7^ pfu of VEE-HA provided complete protection in chickens. RNA-based immunization with 10 µg of SFV-HA RNA replicons elicited significant immune responses in BALB/c mice and provided protection in 90% of vaccinated animals [53]. In comparison to conventional mRNA immunization, only 1.25 µg of self-amplifying VEE-HA RNA was required to acquire protection against challenges with influenza A virus H1N1, H3N2 and B strains compared to 80 µg of synthetic mRNA [54]. Moreover, a replication-deficient modified vaccinia virus Ankara (MVA) expressing the *HA* gene from influenza virus A/HK/156/97 protected C57BL/6J mice from challenges with the three antigenically distinct strains A/HK/156/97, A/Vietnam/1194/04 and A/Indonesia/5/05 [55]. Most of the clinical development and approvals of influenza vaccines have relied on live-attenuated vaccines. However, limited clinical trials have been conducted with viral vector-based vaccines such as in a phase I/IIa study in 79 healthy volunteers receiving MVA-HA [56]. The vaccination was safe and induced significantly higher antibody titers in individuals receiving a higher dose of 1 × 10^8^ pfu compared to 1 × 10^7^ pfu.

HIV/AIDS has had a substantial impact globally, which has contributed to accelerated efforts to develop vaccines against HIV. Cytomegalovirus has the potential as an attractive candidate for vaccine development due to its feature of systematic induction and maintenance of high levels of effector memory T cells through the “memory inflation” mechanism [57]. This has also included applications of CMV for vaccine development against HIV, as T cell vaccines inducing noncanonical CD8^+^ T cell responses could induce population-wide immunity against HIV [58]. Related to Ad-based HIV vaccine development, it was demonstrated that replication-deficient Ad5 expressing HIV Gag elicited consistently strong, long-lived CD8^+^ biased T cell responses in immunized baboons [59]. In the case of MV, live-attenuated MV expressing HIV-1 Gag like particles with a gp160DeltaV1V2 Env protein envelope elicited high levels of cellular and humoral activity against both MV and HIV with neutralizing activity in immunized mice [60]. Alphavirus vectors have also been subjected to HIV vaccine development, and for instance SFV-HIV-Env particles were compared to vaccines based on a DNA plasmid and a recombinant Env protein [61]. Immunized mice showed the highest antibody titers for the SFV particle-based vaccine. In another study, mice intramuscularly immunized with SFV replicon RNA expressing the HIV-1 Env gene elicited Env-specific antibody responses in four out of five mice [62]. In another approach, recombinant SFV particles and replicon RNA were compared for the expression of the Indian HIV-1C *Env/Gag/PolRT* genes in mice [63]. Significant T cell responses were detected for both particle- and RNA-based immunizations, although the titers were superior for SFV particles compared to RNA. Layered SFV DNA/RNA plasmid vectors expressing HIV Env and a Gag/Pol/Nef fusion protein have also been subjected to immunization studies in BALB/c mice, resulting in strong immune responses [64]. Moreover, alphavirus RNA replicons have been subjected to formulations with a cationic nanoemulsion (CNE) and compared to replicon particles and HIV Env formulated with MF59 adjuvant [65]. The replicon-vector, based on VEE included the HIV-1 glycoprotein 140 (gp140) and the packaging signal of Sindbis virus (SIN) and 3′ end untranslated region, was encapsulated in a CNE consisting of squalene, 1,2-dioleoyl-3-trimethylammonium-propane (DOTAP) and sorbitan trioleate. Intramuscular injection of 50 µg of encapsulated replicon RNA generated potent cellular immune responses in rhesus macaques, which were stronger than immunization with VEE particles or HIV gp140. Moreover, immunization of RNA replicons expressing the HIV glycoprotein 120 (gp120) and encapsulated in DOTAP-based lipid nanoparticles showed higher levels of HIV gp120 expression compared to modified conventional mRNA for 30 days in mice after intramuscular administration [66]. In the context of clinical trials of viral vector-based HIV vaccines, the Ad vaccine failed to show protection against infection in the STEP trial [67]. The vaccine, consisting of three Ad5 vectors expressing the HIV *Gag*, *Pol* and *Nef* genes, respectively, was administered to almost 3000 uninfected volunteers. Of even greater concern was the finding that the vaccine appeared to increase HIV infection rates in individuals with pre-existing immunity against Ad5, which resulted in the premature termination of the trial [68]. For this reason, other vaccine approaches such as DNA prime immunization followed by poxvirus boosting have been subjected to clinical trials [69]. Furthermore, a phase III clinical trial was conducted in Thailand with the Canarypox virus HIV vaccine (ALVAC), based on a canarypox virus, and the AIDSVAX B/E gp120 protein vaccine [70]. Vaccination of 16,402 subjects suggested a trend towards the prevention of HIV infection, but the efficacy was modest, only 32%. Another phase III HIV clinical trial, HVTN 702, in South Africa based on the ALVAC/gp120 vaccine was recently terminated by the National Institute of Allergy and Infectious Diseases following recommendations from an independent data and safety monitoring board indicating that the prime-boost vaccine was not efficacious at preventing HIV [71]. In addition, LV vectors have also been applied for prevention and treatment of HIV showing a high degree of immunogenicity in preclinical studies [72]. Moreover, a LV-based dendritic cell (DC) vaccine expressing the CD40 ligand (CD40L) and the HIV-1 SL9 epitope induced antigen-specific T cell proliferation and memory differentiation in humanized mice [73]. The viral load was reduced significantly (by 2 logs) in immunized mice challenged with HIV-1 and the antiviral response was superior when full-length HIV-1 proteins were expressed from the LV vector (Table 1).

Related to non-viral pathogens, Ad and alphavirus vectors have been applied for vaccine development. For instance, immunization of BALB/c mice with an SFV DNA replicon vector expressing the *Clostridium botulinum* neurotoxin A elicited antibody and lymphoproliferative responses [74]. Moreover, a single intranasal inoculation of a replication-deficient Ad vector expressing the heavy chain C-fragment of the *C. botulinum* neurotoxin C (BoNT/C) elicited high levels of BoNT/C-specific antibodies and protected against challenges with BoNT/C [75]. Related to malaria, SFV particles expressing the *Plasmodium falciparum* Pf332 antigen elicited strong immune responses and immunological memory [76]. Moreover, Ad5- and Ad35-based expression of the *P. falciparum* circumsporozoite surface protein (CSP) elicited both cellular and serologic CSP antigen-specific responses in mice and induced strong malaria-specific immunity [77]. In another study, recombinant SIN particles were applied for the expression of the *P. voelii* circumsporozoite protein (CS), which induced a strong epitope-specific T cell response and provided a high degree of protection against malaria infection in mice immunized with 1 × 10^8^ pfu SIN particles [78]. SIN DNA replicons have also been utilized for the expression of *Mycobacterium tuberculosis* antigen 85A (Ag85A), which provided long-term protection against *M. tuberculosis* in mice immunized with 5 µg SIN DNA [79]. Similarly, immunization of Swiss Webster mice with 1 × 10^7^ pfu SIN particles expressing the protective antigen (PA) for *Bacillus antracis* elicited specific and neutralizing antibodies resulting in partial protection against *B. antracis* challenges [80].

## 3. Viral Vaccines for Cancer

A large number of cancer vaccine studies have been conducted with various viral vectors, as presented by examples below and in Table 2. For instance, glioblastomas have been targeted by SFV particles expressing endostatin [81]. In comparison to SFV-Lac Z particles and RV-based endostatin delivery, SFV-Endostatin showed superior inhibition of tumor growth and reduced intratumoral vascularization in a mouse B16 glioblastoma model. In another study, mice carrying B16 brain tumors were intratumorally administered DCs transduced with SFV-IL-18 particles in combination with IL-12 protein, which enhanced T helper type 1 responses from tumor specific CD4^+^ and CD8^+^ T cells and natural killers and antitumor immunity [82]. In a gene silencing approach, the miRT124 micro-RNA sequences targeting neurons were introduced into the replication-competent SFV4 vector, changing its tropism to mouse glioblastoma cells and following a single intraperitoneal injection into C57BL/6 mice with implanted CT-2A orthotopic gliomas resulted in significant inhibition of tumor growth and prolonged survival [83]. The chimeric VSVΔG-CHIKV vector, where the VSV G protein was replaced by the CHIKV envelope proteins (E3-E2-6K-E1), showed selective infection and elimination of tumor cells with an extended survival of mice with implanted CT-2A tumors from 40 to 100 days [84]. Oncolytic MV vectors expressing green fluorescence protein (GFP), carcinoembryonic antigen (CEA) and sodium iodide symporter (NIS) have demonstrated viral replication and cytopathic effects in glioblastoma cell lines [85]. Moreover, significant antitumor activity was detected in vivo. In a comparative study, Ad5/35 and HSV-1 both demonstrated 70% transduction efficiency in glioma cells [86]. However, in a glioblastoma mouse model where the MV fusogenic membrane glycoprotein (FMG) was expressed from both vectors, HSV-1-based treatment was superior to Ad5/35 therapy. Moreover, the better packaging capacity of HSV-1 favors its future use. In the case of clinical trials, a phase I, dose-escalation study was conducted with the Ad vector DNX-2401 (Delta-24-RGD) in 37 patients with recurrent high-grade glioma (HGG), which resulted in 20% of patients surviving more than 3 years [87]. Additionally, a more than 95% reduction in the tumor size was detected in three patients resulting in over 3 years of progression-free survival.

Related to breast cancer, an Ad vector was engineered with an E2F-1 promoter and the human interleukin-15 (IL-15) gene [88]. The novel SG400-E2F/IL-15 vector selectively killed tumor cells and IL-15 exhibited an immunomodulatory effect, which was confirmed in MDA-MB-231 breast cancer cells. Moreover, strong tumor growth inhibition was observed in BALB/c mice with implanted MDA-MB-231 tumors. Another approach relates to the utilization of adeno-associated virus (AAV) vectors for the delivery of short hairpin RNA (shRNA) targeting basal-like breast cancer (BLBC) [89]. It was demonstrated that the rAAV-PSMA2-shRNA vector efficiently transduced the BLBC cell lines, MDA-MB-468 and HCC1954, resulting in significantly decreased cell viability and induced apoptosis. Moreover, administration of rAAV-PSMA2-shRNA to a BLBC xenograft mouse model resulted in reduced tumor growth. In another AAV-based strategy, delivery of heart-specific miRNA sequences (miRT-1d) supported tumor-specific transgene expression and almost complete elimination in heart tissue [90]. Furthermore, insertion of the therapeutic suicide gene HSV-TK showed significant inhibition of tumor growth in polyoma middle T transgenic mice with multifocal breast tumors. In the context of breast cancer, Ad particles and a SIN DNA replicon expressing the rat HER2/neu gene showed inhibition of A2L2 tumor growth in pre-immunized BALB/c mice but not when the vaccination took place two days after the tumor challenge [91]. A prime-boost regimen with SIN DNA and Ad particles resulted in significant prolongation of survival rates. Moreover, it was demonstrated that intradermal immunization with SIN-HER2/neu DNA replicons elicited strong antibody responses in BALB/c mice [92]. Tumor protection was achieved with 80% less replicon DNA compared to conventional DNA plasmid vectors. In another approach the coxsackievirus A21 (CVA21) was applied for the expression of intercellular adhesion molecule-1 (ICAM-1) and decay-accelerating factor (DAF) [93]. Intravenous injection of CVA21-ICAM-1-DAF combined with intraperitoneal administration of doxorubicin hydrochloride resulted in significantly enhanced tumor regression in mice with MDA-MB-231 breast tumors. Related to clinical trials, six patients with recurrent breast cancer were included in a phase I dose-escalation study with an oncolytic HSV HF10 vector [94]. The outcome was no serious adverse events, and some therapeutic efficacy was registered.

In the case of cervical cancer, alphaviruses have been frequently used for preclinical immunization studies. For instance, VEE particles expressing the human papilloma virus-16 (HPV-16) E7 protein elicited CD8^+^ T cell responses and prevented tumor development in immunized C57BL76 mice [95]. Moreover, when the HPV E6-E7 fusion was expressed from an SFV vector containing the translation enhancer signal from the SFV capsid gene, immunization of mice with SFVenh-HPV E6-E7 particles provided tumor regression and complete eradication of established tumors [96]. In another study, the combination of intradermal administration of SFV-HPV E6-E7 DNA replicons and electroporation resulted in 85% of immunized mice becoming tumor-free [97]. Remarkably, the therapeutic efficacy was achieved with a 200-fold lower dose, equivalent to 0.05 µg of SFV DNA, compared to conventional DNA plasmid vectors. Recently, GMP-grade production of SFV-HPV E6-E7 (Vvax001) has been produced for use in clinical trials [98]. A number of clinical trials have been conducted on HPV vaccines [99]. For instance, a vaccinia virus vector expressing HPV-16/18 E6/7 induced HPV-specific CTL immune responses in 28% and two out of eight patients showed tumor-free condition at 15 and 21 months, respectively, in a phase I/II trial [100]. In a phase III study in patients with HPV-induced anogenital intraepithelial neoplasia (AGIN), immunization with a recombinant MVA encoding the E2 protein from bovine papilloma virus (BPV) resulted in 90% lesion clearance in treated females and in 100% in male patients [101].

In the case of colon cancer, CT26 colon tumor models have been frequently evaluated. For instance, the non-cytopathic KUN vector expressing the granulocyte macrophage-stimulating factor (GM-CSF)—when administered intratumorally to BALB/c mice with CT26 xenografts—induced CD8^+^ T cell responses, resulted in tumor regression and in cure of more than 50% of immunized animals [102]. In another study SFV particles expressing the vascular endothelial growth factor receptor-2 (VEGFR-2) was used for the immunization of BALB/c mice resulting in inhibition of tumor growth, reduction in tumor angiogenesis and prevention of metastatic spread [103]. Combination therapy with SFV-VEGFR-2 and SFV-IL-12 particles showed lower immune responses and inferior tumor growth inhibition compared to SFV-VEGFR-2 and SFV-IL-4 co-administration, which enhanced VEGFR-2-specific antibody responses and resulted in prolonged survival of immunized mice. Furthermore, immunization of mice with SFV-LacZ RNA replicons elicited antigen-specific and CD8^+^ T cell responses after a single injection of 0.1 µg RNA [104]. Protection against tumor challenges was also achieved and tumor regression was observed in mice with pre-existing tumors. The vaccinia virus cowpox virus (CPVX) was engineered for improved tumor selectivity and oncolytic activity by the introduction of the fusion suicide gene-1 (FCU1), which converts the non-toxic prodrug 5-fluorocytosine (5-FC) into cytotoxic 5-fluorouracil (5-FU) and 5-fluorouridine-5′-monophosphate (5-FUMP) [105]. Systemic administration of the modified CPVX vector showed low accumulation in normal tissues but high tumor selectivity, which induced relevant inhibition of tumor growth. Moreover, co-administration of 5-FC enhanced the anti-tumor effect. Intratumoral CPVX administration induced relevant tumor growth inhibition in a LoVo colon cancer model. An Ad vector expressing CEA was administered to a mouse MC-38 colon cancer model [106]. Immunization of Ad-CEA in combination with the anti-PD-1 antibody showed enhanced anti-tumor activity and immune responses. Related to clinical trials, an oncolytic vaccinia virus was subjected to a phase I study in 11 patients with refractory advanced colorectal or other solid cancers [107]. No dose-related toxicity or treatment-related severe adverse events were detected and a strong inflammatory and Th1 cytokine induction support the potential immunity against cancer. The Newcastle disease virus (NDV) was subjected to immunotherapy in a phase III trial in 335 colorectal cancer patients [108]. The study indicated that NDV vaccinations provided prolonged survival and short-term improvement in quality of life.

Lung cancer has been targeted by alphavirus vectors and SFV-EGFP particles induced cell death in human H358a non-small cell lung cancer (NSCLC) cells and inhibited growth of H358a spheroids [109]. Intratumoral administration of SFV-EGFP to nu/nu mice with H358a xenografts induced apoptosis, which generated complete tumor regression in three out of seven mice. In another study, replication-competent SFV (VA7)-EGFP particles were compared to a conditionally replicating Ad vector (Ad5-Delta24TK-GFP) in nude mice implanted with A549 adenocarcinoma lung cells, which resulted in superior survival of SFV-immunized mice [110]. In contrast, systemic administration did not generate significant immune responses. In another study, immunization with SIN-LacZ particles elicited long-lasting memory T cell responses and provided protection against tumor challenges in mice [111]. Moreover, nude mice immunized with H2009 and A549 lung tumors showed reduced tumor growth after intratumoral administration of VSV-IFNβ [112]. Additionally, intratumoral injection of VSV-IFNβ resulted in tumor regression, extended survival, and the cure of 30% of mice with syngeneic LM2 lung tumors. In another approach, the Edmonston strain of MV expressing CEA showed potent killing of lung cancer cell lines and tumor regression in immunized mice [113]. Related to clinical trials, 78 NSCLC patients were treated with the TG4010 vaccine based on the MVA strain expressing human mucin-1 (MUC-1) and IL-2 [114]. It was discovered that improvement in survival correlated with the development of T cell responses against MUC-1.

Viral vector-based melanoma vaccine research has been intense with numerous preclinical studies and clinical trials conducted. In addition to the parallel study on colon cancer and melanoma for KUN-GM-CSF described above [102], yellow fever virus (YFV)—expressing the CTL epitope SIINFEKL of chicken ovalbumin—elicited SIINFEKL-specific CD8^+^ lymphocytes and protected mice against challenges with malignant melanoma cells [115]. Alphaviruses have been frequently employed for melanoma treatment, where VEE vectors expressing the tyrosine-related protein-2 (TRP-2) demonstrated humoral immune responses, strong antitumor activity, and prolonged survival in a B16 mouse melanoma model [116]. Combination therapy with anti-CTL antigen-4 (CTLA-4) and anti-glucocorticoid-induced tumor necrosis factor receptor (GITR) monoclonal antibodies (mAbs) resulted in complete tumor regression in 50% and 90% of mice, respectively [117]. In another approach, co-administration of SFV-based expression of VEGFR-2 and IL-12 from one DNA replicon and survivin and β-hCG antigens from another DNA replicon was evaluated in a B16 mouse melanoma model [118]. Superior tumor growth inhibition and prolonged survival was achieved by combination therapy in comparison to immunization with either SFV DNA replicon alone. Moreover, the MV Leningrad-16 (L-16) strain showed statistically significant inhibition of tumor growth in a mel Z mouse melanoma model [119]. In another study, the VSV-GP vector pseudotyped with the non-neurotropic lymphocytic choriomeningitis virus (LCMV) provided prolonged survival in mice A375 xenograft and B16-OVA syngeneic mouse models [120]. Application of NDV vectors expressing IL-15 or IL-12 for intratumoral immunization of B16F10 melanoma tumor-bearing mice effectively suppressed tumor growth [121]. The 120-day survival rate for mice treated with rNDV-IL15 was 12.5% higher than for rNDV-IL12. Moreover, tumor re-challenge experiments indicated that the survival rate was 26.7% higher for rNDV-IL15 compared to rNDV-IL12. Furthermore, replication-competent CVA21 expressing ICAM-1/DAF resulted in rapid suppression of subcutaneous SK-Mel-28 melanoma xenografts in NOD-SCID mice [122]. Several clinical trials have been conducted for viral-based melanoma vaccines [123]. In this context, HSV-1 has been subjected to clinical phase I/IIb and phase III trials [124,125]. In the former, a 50% objective response rate was obtained in patients and a durable response lasting for more than 6 months was seen in 44% of patients [124]. In the phase III trial, the durable response rate improved, and longer median survival rates were obtained in patients with non-surgically resectable melanoma [125]. Moreover, a phase II/IIIb study resulted in significant clinical benefits and superior overall survival in stage III and IV melanoma patients [126]. The replication-competent reovirus, a dsRNA virus, was subjected to a phase II trial in patients with metastatic melanoma [127]. The treatment was well tolerated and reovirus replication was demonstrated in patient biopsies. In the case of CVA21-based clinical trials, stable disease was observed in 26.7% of patients in a phase Ib study [128] and durable responses in melanoma metastases were detected in a phase II trial [129,130]. A 15-year follow-up of an NDV-based clinical phase II trial demonstrated that NDV oncolysates were associated with prolonged survival in patients with lymph node-positive malignant melanoma [131].

Several types of vectors have been applied for vaccine development against ovarian cancer. The pseudotyped VSV-LCMV-GP demonstrated oncolytic activity in several ovarian cancer cell lines and in vivo in an ovarian A2780 tumor mouse model [132]. Superior reduction in tumor size was observed in combination with the JAK1/2 inhibitor ruxolitinib in both subcutaneous and orthotopic xenograft mouse models. Moreover, an MV containing a single-chain antibody (scFv) specific for the alpha-folate receptor (αFR), provided tumor specific targeting with no background infectivity of normal cells [133]. Mice with SKOV3ip.1 xenografts were intratumorally injected with MV-GFP and MV-αFR, which resulted in tumor volume reduction and increase in overall survival. Studies involving alphaviruses have been conducted on ovarian cancer such as combination therapy of SIN-IL-12 particles and the CPT-11 topoisomerase inhibitor irinotecan, which provided long-term survival in SCID mice implanted with aggressively growing human ovarian ES2 tumors [134]. Additionally, a prime-boost regimen of SFV-OVA and VV-OVA resulted in enhanced OVA-specific CD8+ T cell immune responses and enhanced anti-tumor activity in immunized C57BL/6 mice with implanted murine ovarian surface epithelial carcinoma (MOSEC) [135]. Related to clinical trials, a phase I study in patients with stage II-IV ovarian epithelial, fallopian tube, or primary peritoneal cavity cancer with the poxvirus ALVAC is in progress [136]. In a similar phase I trial, the safety and tolerability of the ALVAC vaccine was determined [137]. Furthermore, a phase II trial with fowlpox vaccinia virus in patients with epithelial ovarian, fallopian tube, or primary peritoneal carcinoma and whose tumors expressed the NY-ESO-1 or LAGE-1 antigen, evaluated the maintenance of remission at 12 months, time to failure of vaccine therapy, and cellular and humoral immunity [138].

In the case of pancreatic cancer, AAV2 expressing endostatin was administered intramuscularly or intravenously (portal vein) into Syrian golden hamsters previously inoculated into the pancreas with PGHAM-1 pancreatic cells [139]. The transplanted PGHAM-1 cells rapidly metastasized to the liver. After intramuscular injection the endostatin levels showed a modest increase and the numbers of metastases decreased. Intraportal administration resulted in significantly increased levels of endostatin and the size and number of metastases decreased substantially. Overall, intraportal injection was more efficient as an anti-angiogenic therapy. Oncolytic Ad vectors engineered with cell-targeting ligand SYENFSA (SYE) have demonstrated specific targeting of pancreatic cancer cells and efficient oncolysis of pancreatic ductal adenocarcinoma (PDAC) cells [140]. Moreover, VSV-GFP showed superior oncolytic activity in PDAC cell lines and in vivo compared to a conditionally replicative Ad vector (CRAd), Sendai virus and respiratory syncytial virus (RSV) [140]. However, the pancreatic HPAF-II cell line and a mouse HPAF-II model were resistant to VSV infections, which could be reduced by combination therapy with DEAE-dextran and ruxolitinib [141]. In another approach, SCID mice implanted with KLM1 and Capan-2 xenografts were immunized with MV vectors expressing SLAMblind showing significant suppression of tumor growth [142]. Furthermore, the chimeric orthopoxvirus CF33 efficiently killed six pancreatic cancer cell lines and caused regression in PANC-1 pancreatic xenografts after a single intratumoral injection of a low dose of 10^3^ pfu [143]. CF33 was shown to preferentially replicate in tumors and non-injected distant xenografts were also affected. Related to clinical trials, eight patients with nonresectable pancreatic cancer were immunized intratumorally with an oncolytic HSV HF10 vaccine in a phase I dose-escalation study [94]. No serious adverse events occurred, and therapeutic efficacy was registered. In another phase I study, patients with nonresectable locally advanced pancreatic cancer showed only HSV HF10-unrelated adverse events after intratumoral administration [144]. Three patients showed partial responses (PR), stable disease (SD) was observed in four patients, and nine patients had progressive disease (PD). VEE-CEA vectors were administered intramuscularly in a phase I trial in pancreatic cancer patients [145]. Repeated VEE-CEA administration induced clinically relevant CEA-specific T cell antibody responses.

In the context of prostate cancer, intratumoral immunization with MV-CEA vectors resulted in a significant delay of tumor growth and prolonged survival in a prostate PC-3 mouse model [146]. Application of alphaviruses has demonstrated strong specific immune responses against prostate-specific membrane antigen (PSMA) [147] and six-transmembrane epithelial antigen of the prostate (STEAP) [148] after immunization of mice with VEE-PSMA and VEE-STEAP, respectively. Transgenic adenocarcinoma mouse prostate (TRAMP) mice showed long-term survival of 90% at 12 months after immunization with VEE particles expressing the prostate stem cell antigen (PSCA) [149]. VSV-LCMV-GP expressing luciferase (Luc) efficiently infected prostate cancer cell lines and showed long-term remission in intratumorally immunized Du145 and 22Rv1 mouse prostate cancer models [150]. In another approach, the combination therapy of oncolytic MV and mumps virus (MuV) vectors showed greater antitumor activity and prolonged survival in a PC-3 human prostate cancer model in comparison to MV and MuV vectors alone [151]. Related to clinical trials, in a phase I study, patients with castration resistant metastatic prostate cancer (CRPC) were immunized with either 0.9 × 10^7^ or 3.6 × 10^7^ IU of VEE-PSMA [152]. The treatment was well tolerated, but induced only weak PSMA-specific immune responses, which will require dose optimization to enhance the efficacy. In a phase I trial in 32 patients with hormone refractory metastatic prostate cancer vaccination with Ad5 expressing prostate-specific antigen (PSA), anti-PSA antibodies were elicited in 34% of patients, 68% showed anti-PSA responses, 48% had a longer PSA doubling time, and the survival time was prolonged in 55% of the patients [153]. POSTVAC (TRICOM) is a poxvirus vaccine candidate based on an attenuated recombinant VV prime vector and a fowlpox virus booster vector expressing B7-1, lymphocyte function associated antigen-3 (LFA-3) and ICAM-1 [154]. In a phase II trial, 125 minimally symptomatic CRPC patients were immunized with PROSTVAC, which showed an increase in the median overall survival but not progression free survival [155]. Similar findings were obtained in another phase II trial in 32 CRPC patients treated with PROSTVAC and GM-CSF [156]. Furthermore, in a phase III study in GRPC patients no differences were found in overall survival between patients treated with PROSTVAC, PROSTVAC + GM-CSF or placebo [157]. These findings indicate that other combination therapies with DNA vaccines and chemotherapies need to be explored [158] (Table 2).

## 4. Vaccines against COVID-19

Naturally, vaccine development against the severe acute respiratory syndrome-coronavirus-2 (SARS-CoV-2) causing the COVID-19 pandemic has overshadowed any other vaccine initiative [159]. The impressive number of 155 vaccine candidates in preclinical and 47 candidates in clinical trials are based on inactivated and live attenuated vaccines, protein subunit and peptide vaccines, nucleic acids and viral vectors [160]. The focus here is uniquely on viral vector-based vaccines (Table 3).

The chimpanzee Ad vector ChAdOx1 nCoV-19 was engineered to express the SARS-CoV-2 S protein and when subjected to immunization of mice and rhesus macaques induced strong humoral and cellular immune responses and prevented pneumonia in macaques [161,162]. Similarly, Ad5-SARS-COV-2 S elicited strong S-specific antibody and cell-mediated immune responses in mice and rhesus macaques. Moreover, a single intramuscular or intranasal immunization with Ad5-S-nb2 provided protection against challenges with SARS-CoV-2 in macaques [163]. Preclinical studies in hamsters demonstrated that a single immunization with an Ad26 vector expressing SARS-CoV-2 S elicited neutralizing antibodies and protected immunized animals against pneumonia and death [164]. Immunization of rhesus macaques elicited strong neutralizing antibody responses and protected primates against SARS-CoV-2 [165]. In another preclinical approach, the full-length SARS-CoV-2 S gene was inserted into two positions of the MV genome [166]. Administration of the vaccine candidates to mice demonstrated efficient Th1-biased antibody and T cell responses after two immunizations. Considering that the lung is a vital organ for SARS-CoV-2 infection, MVA poxviruses have been suggested as potential candidates for COVID-19 vaccine development [167]. In this context, a novel vaccine platform was developed for MVA, where a unique three-plasmid system can efficiently generate recombinant MVA vectors from chemically synthesized DNA [168]. Using this technology, mice were immunized with fully synthetic MVA (sMVA) vectors co-expressing SARS-CoV-2 S and nucleocapsid, which elicited robust SARS-CoV-2 antigen-specific humoral and cellular immune responses including potent neutralizing antibodies.

Positive results from preclinical studies on COVID-19 vaccine candidates have supported the launch of several clinical trials. The first-in-human phase I dose-escalation, non-randomized clinical trial was conducted with three doses (5 × 10^10^, 1 × 10^11^ and 1.5 × 10^11^) of Ad5-SARS-CoV-2 S particles in 108 healthy volunteers [169]. The safety and tolerability of the treatment was good with only some minor pain reactions to the vaccination. Rapid SARS-CoV-2-specific T cell responses were detected 14 days after vaccination and humoral responses against SARS-CoV-2 reached peak levels at day 28 post-immunization. The Ad5-SARS-CoV-2 S vaccine candidate has now been subjected to a randomized, double-blind, placebo-controlled phase II trial in 603 healthy volunteers [170]. The two doses (1 × 10^11^ and 5 × 10^10^ virus particles) elicited significant neutralizing antibodies. Severe adverse reactions were observed in 24 (9%) of vaccinees, but no serious adverse reactions were reported. Overall, the immunization was safe and significant immune responses were induced in the majority of vaccinees after a single vaccination. Moreover, the recruitment of healthy adults 18 years of age and older is in progress for a global double-blind, placebo-controlled phase III trial with an immunization schedule of one intramuscular dose of Ad5-SARS-CoV-2 S [171]. Recruitment is in progress for a similar phase III trial for 18 to 85 years old volunteers for a single intramuscular administration of Ad5-SARS-CoV-2 S [172]. In another Ad based approach, the Ad26.COV2-S vaccine candidate was subjected to a randomized, double blind, placebo-controlled phase I/II study in 1045 healthy volunteers in Belgium and the USA [173]. Interim results demonstrated a good safety profile and immunogenicity after a single immunization [174]. A randomized, double-blind, placebo-controlled phase III study enrolling 60,000 participants is in progress [175].

The Ad-based Sputnik V vaccine developed at the Gamaleya Research Institute of Epidemiology and Microbiology in Russia caused some controversy due to its premature approval prior to the completion of any clinical phase III trials and even before the publication of findings from any preclinical or clinical studies with only a preliminary evaluation in 76 volunteers [176]. The rAd26-S/rAd5-S vaccine regimen is based on a prime vaccination with the Ad26-based SARS-CoV-2 S, followed by a booster vaccination with Ad5-SARS-Cov-2 S. Several weeks after the approval, the results from a phase I/II trial were published [177]. The results indicated a good safety profile with only mild and no serious adverse events. The intramuscular administration elicited strong SARS-CoV-2-specific antibodies in all vaccinated individuals. Despite being approved weeks earlier, the following statement was made in the publication: “further investigation is needed of the effectiveness of this vaccine for prevention of COVID-19” [177]. Recently, recruiting for two randomized, double-blind, placebo-controlled, multi-center phase III clinical trials in adult volunteers has started [178,179]. The simian ChAdOx1 nCoV-19 vaccine candidate showed promising preliminary results in a phase I/II trial [180]. The safety was good with no serious adverse events registered after a single intramuscular injection. The immune response was also promising with 32 out 35 vaccinees generating SARS-CoV-2-specific neutralizing antibodies. After a booster immunization, both humoral and cellular immune responses were detected in all vaccinees. The ChAdOx1 nCoV-19 vaccine candidate entered a randomized, double-blind, placebo-controlled multicenter phase III trial in 30,000 adults in August 2020 [181]. However, due to some suspect adverse events in patients, the phase III trial was put on hold in early September [182]. After an investigation into the issue, the trial resumed in the UK, but it remained on hold in the US until the FDA authorized the restart on 23 October 2020 [183].

Recently, the first-in-human phase I clinical trial with the MVA-SARS-2-S vaccine candidate in healthy volunteers was approved [184]. The study aims at assessing the safety and tolerability of the vaccine candidate and the enrolment of patients is in progress. A LV vector vaccine candidate based on minigenes of multiple conserved regions of SARS-CoV-2 is planned for a phase I/II clinical trial in 100 healthy volunteers [185]. Subcutaneous administration of 5 × 10^6^ dendritic cells (DCs) transduced with the LV vector (LV-DC) in combination with intravenously injected 1 × 10^8^ antigen-specific CTLs will be evaluated for safety and immunogenicity. Very recently, the MV-SARS-CoV-2 vaccine candidate TMV-083 was subjected to a randomized, placebo-controlled, two-center phase I clinical trial to evaluate the safety, tolerability and immunogenicity in 90 volunteers [186]. As it has been previously demonstrated that the replication-competent VSV-based SARS-CoV-2 S vaccine candidate (V590) can protect mice from SARS-CoV-2 pathogenesis [187], a phase I trial on the safety and tolerability is planned for 252 participants [188]. In another approach, a replication-competent VSV-ΔG vaccine, where the VSV G protein was replaced by SARS-CoV-2 S, resulted in potent SARS-CoV-2-specific neutralizing antibody responses in immunized golden Syrian hamsters [189]. Moreover, a single dose of 5 × 106 pfu of VSV-ΔG vaccine provided protection of hamsters against challenges with lethal doses of SARS-CoV-2. Additionally, the lung damage in immunized animals was minor and no viral load was detected. Next, the VSV- ΔG vaccine will be evaluated in humans in two phases [190]. In a phase I dose-escalation study, 18–55 years old volunteers will receive a single dose of 5 × 10^5^, 5 × 10^6^ and 5 × 10^7^ pfu, respectively. In phase II, elderly subjects will receive a single dose as used in phase I or two immunization with 5 × 10^5^ pfu 28 days apart. Finally, intranasal SARS-CoV-2 vaccine delivery is a potential option [191]. For instance, intranasal administration of an Ad5-based vector expressing the SARS-CoV-2 S receptor binding domain (RBD) elicited strong neutralizing antibody responses [192] (Table 3).

## 5. Conclusions

The progress on viral vector-based vaccine development has been steady, targeting both infectious diseases and different types of cancers. Proof-of-concept has been demonstrated in numerous animal models resulting in robust antibody responses and protection against challenges with pathogens and tumor cells. Moreover, findings from vaccine trials have been encouraging. For instance, several vaccine candidates, based on VSV vectors, have provided protection in phase III trials [37,38]. Moreover, the EBOV vaccine based in the VSV-ZEBOV vector was approved in December 2019 under the brand name Ervebo by the FDA [193]. In the case of cancer vaccines, clinical data have confirmed robust immune responses previously shown in preclinical animal tumor models. Moreover, partial responses, stable disease, and prolonged overall survival have been demonstrated in clinical trials. For example, talimogene laherparepvec (TVEC), the oncolytic HSV-1 vector expressing GM-CSF, was approved for treatment of advanced melanoma by the FDA in October 2015 [194].

As presented in this review, there are many viral vectors to choose between for vaccine development. Clearly, not a single vector system can be declared superior. Although packaging capacity of foreign genes can be of importance, both viral antigens and tumor-associated antigens can be easily accommodated in almost any viral vector. Efficient packaging cell line systems have been engineered for many vector systems such as Ad, AAV, flaviviruses and lentiviruses, which has facilitated rapid and efficient large-scale production of vaccine candidates eligible for clinical applications. Self-replicating RNA virus vectors based on alphaviruses, flaviviruses, measles viruses and rhabdoviruses provide highly efficient cytoplasmic RNA amplification, a substantially favorable feature for generation of enhanced immune responses with reduced vaccine doses. In any case, it is not possible to recommend any universal vector system and each case needs to be evaluated based on the vaccine target, the handling of viral vectors and the preferred route of administration. Obviously, dealing with viral vectors requires a special attention related to safety. Since the advent of application of viral vectors, we have come a long way in engineering replication-deficient and oncolytic versions, which have proven safe for administration to humans. In comparison to conventional vaccines, viral vector-based vaccines have proven competitive related to costs and efficacy. In particular, alphavirus-based vaccines delivered as DNA or RNA replicons have been demonstrated to provide similar immune responses in preclinical animal models at 100- to 1000-fold lower concentrations compared to conventional DNA or RNA vaccines [16,54]. Similarly, protection against lethal challenges was obtained with 1 × 10^6^–10^7^ pfu of self-replicating RNA virus particles [20,22] compared to at least 1 × 10^10^ pfu Ad particles required [27,35]. Furthermore, approval of Ervebo and TVEC by the FDA presents strong evidence of the feasibility of additional viral vector-based vaccines reaching the market. However, further optimization related to vector engineering, delivery and dosing is required.

Finally, the COVID-19 pandemic has surely demonstrated how accelerated vaccine development can be realized. Today, 151 vaccine candidates have been subjected to preclinical studies and 42 vaccines have reached clinical trials. Although other approaches such as live-attenuated, peptide-, protein subunit-, DNA- and RNA-based vaccines have been taken, at least two Ad-based vaccine candidates are currently in phase III and one Ad-based vaccine has been approved, although only in Russia so far. COVID-19 vaccine development presents a good example on several levels. It demonstrates that in a time of a global crisis it is possible for academic institutions and commercial entities to work together efficiently. Moreover, the pandemic has demonstrated that it is appropriate and feasible to develop vaccine candidates based on different strategies including various types of viral vectors to achieve the goal as quickly as possible. Based on the current findings from both preclinical studies and clinical trials it is most likely that one type of COVID-19 vaccine will not be sufficient to overcome the pandemic. Therefore, it is of greatest importance that vaccine development can continue on all fronts with innovation and scientific approval as the cornerstone of all activities.

## Figures and Tables

**Table 1 viruses-12-01324-t001:** Examples of preclinical and clinical vaccine studies for infectious diseases.

Target	Antigen	Vector	Response	Reference
**Alphaviruses**				
CHIKV	E3-E2-6K-E1	VSV	Protection against CHIKV in mice	[19]
VEE	E3-E2-6K	Ad	Protection against VEE in mice	[20]
VEE	E3-E2-6K	VEE	Protection against VEE in mice, macaques	[21]
EEE	E3-E2-6K	EEE	Protection against VEE in mice, macaques	[21]
WEE	E3-E2-6K	WEE	Only weak protection in macaques	[21]
VEE	V4020 strain	VEE DNA	Protection against VEE in mice	[23]
VEE	V4020 strain	VEE DNA	Protection against VEE in macaques	[24]
**Arenaviruses**				
LASV	LASV-GPC	VSV	LASV protection in guinea pigs, macaques	[25]
	LASV-GPC	LASV	Protection against LASV in guinea pigs	[26]
	LASV-GPC/NP	Ad5	Protection against LASV in guinea pigs	[27]
	LASV-GPC	MV	Protection against LASV in macaques	[28]
	LASV-GPC	MV	Phase I trial in progress (healthy volunteers)	[29]
JUNV	JUNV-GPC	VEE	Protection against JUNV in guinea pigs	[30]
MACV	MACV-GPC	VEE	Protection against MACV in guinea pigs	[30]
**Filoviruses**				
EBOV	GP/D637L	KUN	Protection against EBOV in 75% of primates	[32]
	EBOV-GP	VSV	Protection against EBOV in macaques	[33,34]
	EBOV-GP	Ad5	Protection against EBOV in primates	[35]
	EBOV-GP	HPIV3	Protection against EBOV in guinea pigs	[36]
	EBOV-GP	VSV	Good protection against EDV in phase III	[37,38]
	MARV-GP	VSV	Protection against MARV in macaques	[34]
	SUDV-GP	VEE	Protection against SUDV in macaques	[39]
**Flaviviruses**				
DENV	E85	VEE	Protection against DENV in mice	[40]
	ED3	MV	Partial protection against DENV in mice	[41]
ZIKV	prME	VEE-NLC	Protection against ZIKV with 10 ng NLC-	[43]
		-RNA	RNA in mice	
	ME	VSV	Protection against ZIKV in mice	[19]
	ME	DENV	Good safety, neutralizing Abs in volunteers	[45]
**Hepatotropic**				
HBV	HBsAg/HBcAg	Ad7	HBV-specific antibody responses in dogs	[46]
	HBsAg	MV	Partial protection against HBV in primates	[47]
	MHB	SFV-G	Protection against HBV challenges in mice	[48]
**Influenza**				
Influenza A	HA	Ad	Complete protection in mice and chickens	[51]
	HA	VEE	Protection in chicken	[52]
	HA	SFV RNA	Protection in chicken	[53]
	HA	VEE RNA	Protection in mice	[54]
	HA	MVA	Protection against 3 IVA strains in mice	[55]
	HA	MVA	High titer antibodies in phase I/II volunteers	[56]
**Lentivirus**				
HIV	HIV Gag	Ad5	Strong T cell responses in baboons	[59]
	HIV gp160 Env	MV	Neutralizing activity in mice	[60]
	HIV Env	SFV	Superior titers to DNA or protein vaccines	[61]
	HIV Env/Gag/Po	SFV	Particle-based response superior to RNA	[63]
	HIV Gag/Pol/Nef	SFV DNA	Strong immune responses in mice	[64]
	HIV TV1 gp140	VEE*RNA-NP	Stronger responses than for VEE, gp140	[65]
	HIV Env gp120	VEE RNA-NP	Superior response to conventional mRNA	[66]
	HIV Gag/Pol/Nef	3 Ad5	Failure to provide HIV protection in phase III,	[68]
			enhanced HIV rate for pre-existing Ad5	
	HIV gp120	ALVAC/gp120Strong T cell responses in baboons	Modest HIV protection of 32% in phase III	[70]
	HIV-1, CD40L	LV-DCs	Reduced viral load in humanized mice	[73]

Ad5, adenovirus type 5; ALVAC, Canarypox virus HIV vaccine; CD40, CD40 ligand; CHIKV, Chikungunya virus; DENV, Dengue virus; E85, ectodomain of DENV envelope protein; EEE, eastern equine encephalitis virus; HA, hemagglutinin; HIV, human immunodeficiency virus; HPIV3, human parainfluenza virus type 3; IVA, Influenza virus A; JUNV, Junin virus; LASV, Lassa virus; LASV-GPC, Lassa virus glycoprotein; LV-CDs, lentivirus-transduced dendritic cells; MACV, Machupo virus; ME, membrane-envelope; MV, measles virus; MVA, modified vaccinia virus Ankara; NLC, nanostructured lipid carrier; SFV, Semliki Forest virus; SIN, Sindbis virus; VEE, Venezuelan equine encephalitis; VEE*, VEE vector with 3′ end untranslated region and packaging signal form SIN; VSV, vesicular stomatitis virus, WEE, western equine encephalitis virus; ZIKV, Zika virus.

**Table 2 viruses-12-01324-t002:** Examples of preclinical and clinical cancer vaccine studies.

Target	Antigen	Vector	Response	Ref
**Brain**				
GBM	Endostatin	SFV	Tumor regression, prolonged survival in mice	[81]
	IL-18 + IL-12	DC-SFV-IL-18	Enhanced antitumor immunity	[82]
CT-2A	miR124	SFV4	SFV replication in tumors, tumor regression	[83]
	Chimeric VLPs	VSVΔG-CHIKV	Tumor targeting, prolonged survival in mice	[84]
GBM	CEA	MV-CEA/GFP	MV replication in tumors	[85]
	MV FMG	Ad5/35	Transduction of glioma cells	[86]
	MV FMG	HSV-1	Superior to Ad in vitro and in vivo	[86]
HGG	oAd	DNX-2401	Long-term survival (>3 years) in phase I	[87]
**Breast**				
MDA-MB231	Ad	Ad-EF2/lL-15	Tumor growth inhibition in vitro, in mice	[88]
BLBC	PSMA2 shRNA	AAV	Reduced tumor growth in mouse model	[89]
MFB	miRT-1d, HSV-tk	AAV	Significant tumor growth inhibition in mice	[90]
A2L2	HER2/neu	Ad/SIN DNA	Tumor growth inhibition in mice	[91]
	HER2/neu	SIN DNA + Ad	Prolongation of survival in mice	[91]
	HER2/neu	SIN DNA	Tumor protection with 80% less DNA	[92]
MDA-MB231	ICAM-1/DAF	CVA21	Strongly enhanced tumor regression in mice	[93]
Recurrent BC	oHSV	HSV HF10	Safety confirmed in phase I trial	[94]
**Cervical**				
C3	HPV E7	VEE	T cell responses, prevention of tumors	[95]
	HPV E6-E7	SFVenh	Complete eradication of established tumors	[96]
TC-1	HPV E6-E7	SFV DNA	85% tumor-free, 200-fold lower DNA dose	[97]
Adv CC	HPV-16/18 E6/7	VV	CTL in 28% of pts, 2 pts tumor-free in phase I	[100]
AGIN	BPV E2	MVA	90–100% lesion clearance in phase III	[101]
**Colon**				
CT26	GM-CSF	KUN	Tumor regression, cure of >50% of mice	[102]
	VEGFR-2	SFV	Inhibition of tumor growth and metastases	[103]
	VEGFR-2/IL-4	SFV	Prolonged survival in mice	[103]
	LacZ	SFV RNA	T cell responses, protection against tumors	[104]
LoVo	FCU1	CPVX	Tumor selectivity, tumor regression in mice	[105]
MC-38	CEA + anti-PD-1	Ad	Enhanced immune and anti-tumor responses	[106]
Phase I	CD	vvDD	Strongly induced immune responses in pts	[107]
Phase III	NDV 73-T	NDV	Prolonged survival in colon cancer patients	[108]
**Lung**				
NSCLC	EGFP	SFV	Complete tumor regression in 3 out of 7 mice	[109]
A549	EGFP	SFV vs. Ad	Superior survival of SFV over Ad therapy	[110]
CT26.CL25	EGFP	SIN	Protection against tumor challenges	[111]
A549, LM2	IFNβ	VSV	Tumor regression, cure of 30% of mice	[112]
A549, H2009	CEA	MV	Tumor regression in mice	[113]
Phase II	MUC-1, IL-2	MVA	T cell responses, improved survival of pts	[114]
**Melanoma**				
B16-OVA	GM-CSF	KUN	T cell responses, tumor regression in mice	[102]
B16-OVA	SIINFEKL	YFV	Protection against malignant melanoma	[115]
B16	TRP-2	VEE	Prolonged survival in mice	[116]
B16	TRP-2 + mAbs*	VEE	Complete tumor regression in 50–90% of mice	[117]
B16	VEGFR-2/IL-12 +	SFV DNA	Superior tumor growth inhibition after combination	[118]
	Survivin/β-hCG		therapy	
mel Z	MV L-16	MV	Inhibition of tumor growth in mice	[119]
A549, B16	GFP, Luc	VSV-LCMV GP	Prolonged survival in mice	[120]
B16F10	IL-15/IL-12	NDV	Efficient suppression of tumor growth	[121]
SK-Mel-28	ICAM-1/DAF	CVA21	Suppression of tumor growth in mice	[122]
Phase I/IIb	GM-CSF	HSV-1 T-VEC	50% objective response rate lasting > 6 months	[124]
Phase III	GM-CSF	HSV-1 T-VEC	Improved response, longer median survival	[125]
Phase II/IIIb	GM-CSF	HSV-1 T-VEC	Superior overall survival at stage III/IV	[126]
Phase II	Reolysin	Reovirus	Well tolerated, reovirus replication in biopsies	[127]
Phase 1b	CAVATAK	CVA21	Stable disease in 26.7% of patients	[128]
Phase II	CAVATAK	CVA21	Durable responses in metastatic melanoma	[129]
Phase II	NDV oncolysate	NDV	Prolonged survival in melanoma patients	[131]
**Ovarian**				
A2780	Luc + Rux	VSV-LCMV GP	Reduction in tumor growth	[132]
SKOV3ip.1	GFP, αFR	MV	Reduced tumor volume, prolonged survival	[133]
ES2	IL-12, CPT-11	SIN + CPT-11	Long-term survival in SCID mice	[134]
MOSEC	OVA	SFV	Enhanced anti-tumor activity in mice	[135]
Phase I	ALVAC	VV	Safety and tolerability studies	[136,137]
Phase II	Fowlpox	VV	Safety, maintenance of remission	[138]
**Pancreatic**				
PGHAM-1	Endostatin	AAV2	Tumor and metastases regression in hamsters	[139]
PADC	SYE	Ad	Efficient oncolysis of PDAC cells	[140]
PANC-1	GFP	VSV	Oncolytic activity in cell lines and in mice	[141]
Su86.86	GFP	VSV	Oncolytic activity in cell lines and in mice	[141]
KLM1,	SLAM	MV	Suppression of tumor growth in mice	[142]
Capan-2	SLAM	MV	Suppression of tumor growth in mice	[142]
PANC-1	Chimeric OPV	CF33	Replication in tumor cells, tumor regression	[143]
Phase I	oHSV	HSV HF10	Safety, therapeutic efficacy	[94]
Phase I	oHSV	HSV HF10	PR and SD in some patients	[144]
Phase I	CEA	VEE	T cell antibody responses	[145]
**Prostate**				
LNCaP	CEA	MV	Prolonged survival in mice	[146]
TRAMP-C	PSMA	VEE	Strong immune response in mice	[147]
TRAMP	STEAP	VEE	Prolonged survival in mice	[148]
TRAMP-PSA	PSCA	VEE	90% survival rate in mice	[149]
Du145, 22Rv1	Luc	VSV-LCMV-GP	Long-term remission in mice	[150]
PC-3	MV, MuV	MV + MuV	Prolonged survival in mice	[151]
Phase I	PSMA	VEE	Modest neutralizing antibodies against PSMA	[152]
Phase I	PSA	Ad5	Antibody responses, prolonged survival	[153]
Phase II	Tricom	PROSTVAC	Prolonged median OS, not PFS	[155]
Phase III	Tricom + GM-CSF	PROSTVAC	Safe, no effect on OS	[157]

AAV, adeno-associated virus; Ad5, adenovirus type 5: Adv CC, advanced cervical cancer; AGIN, anogenital intraepithelial neoplasia; αFR, alpha folate receptor; BLBC, basal-like breast cancer; BPV, bovine papilloma virus; CEA, carcinoembryonic antigen; CD, yeast cytosine deaminase; CVA21, coxsackievirus A21; DAF, decay-accelerating factor; DC, dendritic cell; FCU1, fusion suicide gene 1; GBM. Glioblastoma multiforme; HGG, high-grade glioma; HPV, human papilloma virus; HSV-tk, herpes simplex virus-thymidine kinase; ICAM-1, intercellular adhesion molecule-1; Luc, luciferase; mAbs*, monoclonal antibodies against anti-CTL antigen-4 (CTLA-4) and anti-glucocorticoid-induced tumor necrosis factor receptor (GITR); MFB, multi-focal breast tumor; MOSEC, murine ovarian surface epithelial carcinoma; MV, measles virus; miRT-1d, micro-RNA targeting heart tissue; MVA, modified vaccinia virus Ankara; MuV, mumps virus; NDV, Newcastle disease virus; NSCLC, non-small cell lung cancer; oAd, oncolytic adenovirus; oHSV, oncolytic herpes simplex virus; OVA, ovalbumin; PFS, progression free survival; PROSTVAC, poxvirus vaccine consisting of VV and fowlpox virus; PSA, prostate-specific antigen; PSCA, prostate stem cell antigen; PSMA, prostate specific membrane antigen; pts, patients; Rux, ruxolitinib; SFV, Semliki Forest virus; SFVenh, SFV vector with translation enhancement signal from the SFV capsid gene; shRNA, short hairpin RNA; SIINFEKL, chicken ovalbumin epitope; SIN, Sindbis virus; SLAM, signaling lymphocyte activating molecule; STEAP, six transmembrane epithelial antigen of the prostate; TRAMP, transgenic adenocarcinoma of the mouse prostate; TRICOM, B71, LFA-3 and ICAM-1 expressed from PROSTVAC; VEE, Venezuelan equine encephalitis; VSV, vesicular stomatitis virus, VV, vaccinia virus; vvDD, oncolytic vaccinia virus vector expressing CD; YFV, yellow fever virus.

**Table 3 viruses-12-01324-t003:** Viral vector-based COVID-19 vaccine candidates.

Viral Vector	Stage	Response	Ref
**Adenovirus**			
ChAdOx1 nCoV-19	Preclinical	Strong immune response in mice and macaques	[161]
ChAdOx1 nCoV-19	Preclinical	Prevention of pneumonia in macaques	[162]
ChAdOx1 nCov-19	Phase I/II	Humoral and cellular responses in all vaccinees	[180]
ChAdOx1 nCoV-19	Phase III	Trial on hold because of suspect adverse events	[181]
Ad5-S-nb2	Preclinical	Strong immune response, SARS-CoV-2 protection	[163]
Ad5-S-nb2	Phase I	Humoral and T cell responses in volunteers	[169]
Ad5-S-nb2	Phase II	Significant immune responses in volunteers	[170]
Ad5-S-nb2	Phase III	Recruitment in progress	[171]
Ad5-S-nb2	Phase III	Recruitment in progress	[172]
Ad26.COV2.S	Preclinical	Protection against pneumonia in hamsters	[164]
Ad26.COV2.S	Preclinical	Protection against SARS-CoV-2 in macaques	[165]
Ad26.COV2.S	Phase I/II	Good safety and immunogenicity in volunteers	[173,174]
Ad26.COV2.S	Phase III	Recruitment in progress	[175]
rAd26-S/rAd5-S	Phase I/II	Good safety, humoral and cellular response	[177]
rAd26-S/rAd5-S	Phase III	Recruitment in progress	[178]
rAd26-S/rAd5-S	Phase III	Recruitment in progress	[179]
Ad5-CoV-2 S RBD	Preclinical	Neutralizing antibodies after nasal administration	[192]
**Measles virus**			
MV-SARS-CoV-2 S	Preclinical	Neutralizing and T cell antibody responses in mice	[166]
MV-SARS-CoV-2 S	Phase I	Recruiting in progress	[186]
**Poxviruses**			
sMVA	Preclinical	Potent neutralizing SARS-CoV-2 antibodies in mice	[168]
MVA-SARS-S	Phase I	Recruitment of participants in progress	[184]
**Lentiviruses**			
LV-DCs + CTL Ag	Phase I/II	Safety and immunogenicity evaluations in progress	[185]
**Rhabdoviruses**			
VSV-SARS-CoV2-S	Preclinical	Protection against SARS-CoV-2 pathogenesis in mice	[187]
VSV-SARS-CoV2-S	Phase I	Planned phase I trials on safety and tolerability	[188]
VSV-ΔG	Preclinical	Protection of hamsters against SARS-CoV-2	[189]
VSV-ΔG	Phase I/II	Recruitment in progress	[190]

Ad, adenovirus; Ag, antigen; ChAdOx1-S, simian adenovirus expressing SARS-CoV-2 S protein; CTLs, cytotoxic T lymphocytes; LV-DCs, lentivirus-transduced dendritic cells; MV, measles virus; MVA, modified vaccinia virus Ankara; RBD, receptor binding domain; sMVA, synthetic modified vaccinia virus Ankara; VSV, vesicular stomatitis virus.

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
