# Peer review of "Application of Viral Vectors for Vaccine Development with a Special Emphasis on COVID-19"

_viruses, 2020, doi:10.3390/v12111324_

Round 1
Reviewer 1 Report
Line 141: bot should be both
Line 329: is it possible that some wording got left out by accident?
Author Response
Dear Reviewer,
Thank you for your valuable comments, which have been taken inti account in the revised version as indicated below.
Line 141: bot should be both
R: Correction has been made.
Line 329: is it possible that some wording got left out by accident?
R: Correction has been made.
Reviewer 2 Report
This review describes the development of various viral vectors. It starts with an introduction of viruses used as vectors for vaccines and followed by descriptions of various vaccines against infectious disease, cancer and other diseases in vitro, in vivo in animal models and clinical trials. At the end, it describes the recent development of vaccines against COVID-19. Overall, the review describes large numbers of viral vectors for vaccine developments in largely descriptive way. One suggestion would be to describe rationale of selecting a specific viral vector for a specific disease. What are the advantages, disadvantages, cost and safety of viral vectors compared to conventional vaccine strategies and among virus vectors. What are the challenges for viral vector vaccines to be approved by FDA? It would be more informative if the information are included. Emphasis on COVID-19 vaccines need updates before the publication since the field is moving very rapidly and in particular the results of phase III trials may be available.
Comments
Line 22. Introduction section describes basic features of viruses that have been used for vaccines and the capacity of accommodating a transgene. It would be more informative if the advantages, disadvantages and safety are compared to conventional vaccine strategies and among virus vectors and described in this section.
Line 37 “Briefly, Ad vectors are naked double-stranded DNA (dsDNA) viruses”. The description of “Ad vectors as naked double-stranded DNA viruses is confusing as the virus vectors are not naked. I assume it means non-enveloped.
Author Response
Dear Reviewer,
Thank you for your valuable comments, which have been taken into account in the revised version as indicated below.
One suggestion would be to describe rationale of selecting a specific viral vector for a specific disease. What are the advantages, disadvantages, cost and safety of viral vectors compared to conventional vaccine strategies and among virus vectors. What are the challenges for viral vector vaccines to be approved by FDA?
R: These issues have been addressed by expanding the Conclusions section.
It would be more informative if the information are included. Emphasis on COVID-19 vaccines need updates before the publication since the field is moving very rapidly and in particular the results of phase III trials may be available.
R: The section on COVID-19 vaccines has been updated.
Comments
Line 22. Introduction section describes basic features of viruses that have been used for vaccines and the capacity of accommodating a transgene.
It would be more informative if the advantages, disadvantages and safety are compared to conventional vaccine strategies and among virus vectors and described in this section.
R: Information added on advantages/disadvantages of vectors, comparison to conventional vaccines
Line 37 “Briefly, Ad vectors are naked double-stranded DNA (dsDNA) viruses”. The description of “Ad vectors as naked double-stranded DNA viruses is confusing as the virus vectors are not naked. I assume it means non-enveloped.
R: Correction has been made.
Reviewer 3 Report
This article is cataloging literature surrounding viral vectors for vaccines against both infection and cancer. While tables and the list of references may be useful, very little insight into overview of the technology, virology and immunology behind the R&D, comparisons of different viral vectors for different diseases and future challenges.
To my knowledge, many vectors included as for cancer vaccines that are not encoding either tumour antigens or etiological virus antigens are not really vaccines. Most vectors discussed are primarily oncolytic viruses and only some also have immunogens coded. Vaccinating against GFP for cancer does not make sense to me, while many have GFP as antigen in the table. These should be excluded or referred as something related with some similar effect (e.g. raising/enhancing anti-cancer immunity).
Author Response
Dear Reviewer,
Thank you for your valuable comments, which have been taken into account in the revised version as indicated below.
This article is cataloging literature surrounding viral vectors for vaccines against both infection and cancer. While tables and the list of references may be useful, very little insight into overview of the technology, virology and immunology behind the R&D, comparisons of different viral vectors for different diseases and future challenges.
R: Comparison of viral vectors added to the Conclusions section.
To my knowledge, many vectors included as for cancer vaccines that are not encoding either tumour antigens or etiological virus antigens are not really vaccines. Most vectors discussed are primarily oncolytic viruses and only some also have immunogens coded. Vaccinating against GFP for cancer does not make sense to me, while many have GFP as antigen in the table. These should be excluded or referred as something related with some similar effect (e.g. raising/enhancing anti-cancer immunity).
R: I do acknowledge the point raises by the reviewer. However, I respectfully disagree as the use of oncolytic vectors has been successful in cancer immunotherapy, although not being “purebred” vaccine vectors. The point is not to use GFP or other reporter genes as therapeutic antigens, but they can provide excellent information of delivery efficacy. In any case, this has been pointed out in the text at the end of the Introduction.
Reviewer 4 Report
In this review, the author has put together a fairly comprehensive account of viral vectors and their use as vaccine delivery systems.
I have a few minor suggestions to the paper:
Introduction
- Line 50: when talking about targeted integration of lentiviral vectors, there is no mention of the fact they can be made integration defective and therefore be cleared by the host. This is seen as both a positive and a negative and I think it should be touched upon.
- It is a shame the author has not commented on the lentiviral packaging capacity, despite mentioning this for the other viral vector systems.
Section 2:
- Line 66: Please italicize "in vitro"
- Line 76-77: The sentence in which the author refers to the protection of mice against CHIKV and ZIKV is vague. I think it would pay to add in some additional information about what type of immune response was linked to this protection.
- Line 78: Venezuelan equine encephalitis virus should be abbreviated to VEEV. VEE is for the disease itself (Venezuelan equine encephalitis). Please could the author be consistent with this.
- Line 89: The sentence beginning with 'However' needs re-wording. Perhaps adding 'the' after however and on Line 90, 'seen' after protection.
- Table 1: Please could the author look at the justification of the test. Midway down the table at the section about Flaviviruses, the Vector column falls out of sync with the Response column, although the references do not follow suit.
- Lines 106-109: The sentence beginning with 'Protection' and the following sentence beginning 'Engineering' both need to be re-written as it is not clear the point which is wanting to be put across.
- Line 118: Kunjin virus should be abbreviated KUNV, the V is not present in this paper.
- Line 121: Information on the dose size for each vaccine example given is inconsistent throughout. A dose size should be included here when talking about protection. Also a reference is needed.
- There is no mention of lentivirus examples in this section, despite introducing them in the introduction. Perhaps there should be an example in here too to keep it balanced.
- Line 149, 160, 167: 'In the context of' is used quite a lot in this section (amongst others further on in the paper), please could the author adjust this repetition.
- After Line 170, the topic within this section changes significantly from examples to viral vectors to disease specific examples including multiple vector types. Perhaps the use of a subheading would help make this change of flow more obvious.
Section 3:
- Line 269:Please italicize "in vivo"
- Table 2: please be mindful of text falling out of line. Because of the large amount of data in this table it can be hard to follow. Maybe some inclusion of additional lines would be of benefit.
Section 4:
- Line 552: The ChAdOx1-S the author refers to is actually called ChAdOx1 nCov-19 (AZD1222 is the technical name from AstraZeneca)
- There is no mention of the large animal data on the ChAdOx1 nCoV-19 vaccine comparing single shot to prime boost which was shown to be a good model for SARS-CoV-2 vaccine immunogenicity (https://pubmed.ncbi.nlm.nih.gov/32793398/)
Overall, despite the title suggesting there is a special emphasis on COVID-19, I feel this gets lost a bit in the manuscript due to all the information on Cancer, which there is a plethora of. The use of viral vectors as immunotherapy has been reviewed a lot in the past so maybe this should be accounted for in this review.
Author Response
Dear Reviewer,
Thank you for your valuable comments, which have been taken into account in the revised version as indicated below.
Introduction
- Line 50: when talking about targeted integration of lentiviral vectors, there is no mention of the fact they can be made integration defective and therefore be cleared by the host. This is seen as both a positive and a negative and I think it should be touched upon.
R: Integration-defective lentivirus vectors have been included.
- It is a shame the author has not commented on the lentiviral packaging capacity, despite mentioning this for the other viral vector systems.
R: The requested information has been added.
Section 2:
- Line 66: Please italicize "in vitro"
R: “in vitro* should not be italicized according to the instructions from Viruses
- Line 76-77: The sentence in which the author refers to the protection of mice against CHIKV and ZIKV is vague. I think it would pay to add in some additional information about what type of immune response was linked to this protection.
R: Information added of induction of neutralizing antibodies
- Line 78: Venezuelan equine encephalitis virus should be abbreviated to VEEV. VEE is for the disease itself (Venezuelan equine encephalitis). Please could the author be consistent with this.
R: VEE has been commonly used as the abbreviation for the virus too.
- Line 89: The sentence beginning with 'However' needs re-wording. Perhaps adding 'the' after however and on Line 90, 'seen' after protection.
- Table 1: Please could the author look at the justification of the test. Midway down the table at the section about Flaviviruses, the Vector column falls out of sync with the Response column, although the references do not follow suit.
R: Corrections have been made to Table 1.
- Lines 106-109: The sentence beginning with 'Protection' and the following sentence beginning 'Engineering' both need to be re-written as it is not clear the point which is wanting to be put across.
R: The sentences have been modified accordingly.
- Line 118: Kunjin virus should be abbreviated KUNV, the V is not present in this paper.
R: The “KUN” abbreviation is also commonly used for Kunjin virus.
- Line 121: Information on the dose size for each vaccine example given is inconsistent throughout. A dose size should be included here when talking about protection. Also a reference is needed.
R: Doses for immunizations in various experiments have been included throughout the review.
- There is no mention of lentivirus examples in this section, despite introducing them in the introduction. Perhaps there should be an example in here too to keep it balanced.
R: Applications of lentiviral vectors have been included in the text and in Table 1.
- Line 149, 160, 167: 'In the context of' is used quite a lot in this section (amongst others further on in the paper), please could the author adjust this repetition.
R: The text has been revised to avoid repetition.
- After Line 170, the topic within this section changes significantly from examples to viral vectors to disease specific examples including multiple vector types. Perhaps the use of a subheading would help make this change of flow more obvious.
R: I disagree respectfully as I did not see any differences between this and other sections
Section 3:
- Line 269: Please italicize "in vivo"
R: “in vivo* should not be italicized according to the instructions from Viruses
- Table 2: please be mindful of text falling out of line. Because of the large amount of data in this table it can be hard to follow. Maybe some inclusion of additional lines would be of benefit.
R: Corrections made to Table 2.
Section 4:
- Line 552: The ChAdOx1-S the author refers to is actually called ChAdOx1 nCov-19 (AZD1222 is the technical name from AstraZeneca)
R: The terminology has been revised
- There is no mention of the large animal data on the ChAdOx1 nCoV-19 vaccine comparing single shot to prime boost which was shown to be a good model for SARS-CoV-2 vaccine immunogenicity (https://pubmed.ncbi.nlm.nih.gov/32793398/)
R: Yes, there is, see lines 515-516 in the revised manuscript.
Overall, despite the title suggesting there is a special emphasis on COVID-19, I feel this gets lost a bit in the manuscript due to all the information on Cancer, which there is a plethora of. The use of viral vectors as immunotherapy has been reviewed a lot in the past so maybe this should be accounted for in this review.
R: I feel both viral and cancer therapy/immunization are important parts of the application of viral vectors.
Round 2
Reviewer 3 Report
The author's response to my first round comments are minimal.
I wish the author did more systematic analysis of literature than just adding some brief 'comparison'.
To my knowledge 'cancer vaccine' can be defined differently by different researchers. There is no definition by the author is presented. It may be rather a disservice to the field to present oncolytic virotherapies as 'vaccines' without clarifying 'cancer vaccine' definition.
One error stood out: 'Poxviruses are large dsRNA viruses' I think they have dsDNA genomes.
Author Response
Thank you for pointing out the typo of dsRNA for poxviruses, which now has been corrected to dsDNA.
In the first revision the justification of including oncolytic vectors in the review was explained (Lines 64-67). The explanation has now been expanded and I hope it is at your satisfaction.